# Exogenous Eugenol Alleviates Salt Stress in Tobacco Seedlings by Regulating the Antioxidant System and Hormone Signaling

**DOI:** 10.3390/ijms25126771

**Published:** 2024-06-20

**Authors:** Jiaxin Xu, Tingting Wang, Xiaoyu Wang, Honghao Yan, Peng Liu, Xin Hou, Yun Gao, Long Yang, Li Zhang

**Affiliations:** College of Plant Protection, Shandong Agricultural University, Taian 271000, China; 17866710916@163.com (J.X.); 18653883150@163.com (T.W.); likeol9527@gmail.com (X.W.); 17842443663@163.com (H.Y.); liupeng2003@sdau.edu.cn (P.L.); houxin@sdau.edu.cn (X.H.); gy2040@126.com (Y.G.)

**Keywords:** tobacco, growth and development, antioxidant system, salt tolerance

## Abstract

Salt stress seriously affects crop growth, leading to a decline in crop quality and yield. Application of exogenous substances to improve the salt tolerance of crops and promote their growth under salt stress has become a widespread and effective means. Eugenol is a small molecule of plant origin with medicinal properties such as antibacterial, antiviral, and antioxidant properties. In this study, tobacco seedlings were placed in Hoagland’s solution containing NaCl in the presence or absence of eugenol, and physiological indices related to stress tolerance were measured along with transcriptome sequencing. The results showed that eugenol improved the growth of tobacco seedlings under salt stress. It promoted carbon and nitrogen metabolism, increased the activities of nitrate reductase (NR), sucrose synthase (SS), and glutamine synthetase (GS) by 31.03, 5.80, and 51.06%. It also activated the enzymatic and non-enzymatic antioxidant systems, reduced the accumulation of reactive oxygen species in the tobacco seedlings, and increased the activities of superoxide dismutase (SOD), peroxidase (POD), catalase (CAT), and ascorbate peroxidase (APX) by 24.38%, 18.22%, 21.60%, and 28.8%, respectively. The content of glutathione (GSH) was increased by 29.49%, and the content of superoxide anion (O_2_^−^) and malondialdehyde (MDA) were reduced by 29.83 and 33.86%, respectively. Promoted osmoregulation, the content of Na^+^ decreased by 34.34, K^+^ increased by 41.25%, and starch and soluble sugar increased by 7.72% and 25.42%, respectively. It coordinated hormone signaling in seedlings; the content of abscisic acid (ABA) and gibberellic acid 3 (GA_3_) increased by 51.93% and 266.28%, respectively. The transcriptome data indicated that the differentially expressed genes were mainly enriched in phenylpropanoid biosynthesis, the MAPK signaling pathway, and phytohormone signal transduction pathways. The results of this study revealed the novel role of eugenol in regulating plant resistance and provided a reference for the use of exogenous substances to alleviate salt stress.

## 1. Introduction

Abiotic stresses, such as salinity, low temperature, and drought, have a negative impact on many facets of plant life and are strongly linked to the growth, development, quality, and yield of plants. One major factor limiting plant growth is salt stress [1]. Currently, about 20 percent of the world’s arable land is affected by salinization, and soil salinization is expanding, increasing to 50 percent by 2050, as a result of factors such as irrational irrigation and excessive application of chemical fertilizers in agricultural production [2]. As one of the common environmental factors affecting plant growth, salt stress inhibits normal plant growth, reduces plant productivity, and even kills plants by disrupting the active oxygen balance, osmotic balance, ionic balance, and so on in plants [3]. Salinity has been shown to significantly reduce plant height, stem girth, biomass both above and below ground, and hinder leaf cell division [4]. Additionally, salt stress inhibits root growth [5], reduces root vigor, and its plant photosynthetic process is also greatly affected [6], which destroys chloroplast structure in the plant, increases the number of cysts and basal lamellae [7], and degrades chlorophylls and carotenoids. These changes affect the plant’s ability to carry out photosynthesis, which ultimately affects the plant’s dry matter accumulation and its yield [8]. Reducing the effects of salt stress plays an important role in promoting plant growth and increasing the quality of plant yield.

Salt stress not only impacts the growth and development of plants but also significantly affects their in vivo physiological and biochemical processes. Two crucial metabolic processes in plants, carbon metabolism and nitrogen metabolism, provide the primary energy and essential nutrients for plant life activities and are indispensable for plant growth and development [9]. Nitrogen metabolic activities include nitrogen intake, transport, and amino acid metabolic reduction and assimilation; carbon metabolic processes include photosynthetic carbon assimilation, sucrose and starch metabolism, carbohydrate transport, and utilization [10]. Salt stress is a major abiotic stressor that disrupts the balance of carbon and nitrogen metabolism in plants, thereby hindering plant growth and development. Salt stresses have a significant impact on these metabolic processes [11]. Prior research has demonstrated that salt stress impacts the way that carbohydrates build up in plants and that it also lowers the amount of sucrose and soluble sugars in plants, as well as the activity of the enzymes sucrose phosphate synthase and sucrose synthase [12]. Plants metabolize nitrogen by reducing inorganic forms of NO_3_^−^ to NH_4_^+^ through the glutamine synthase/glutamate synthase cycle, which is then converted to organic forms; however, salt stress led to a decrease in GS activity, and these changes suppressed carbon and nitrogen metabolism in the plant. This was due to a significant reduction in plant nitrate reductase activity [13].

Plant buildup of reactive oxygen species is impacted by salt stress. In plant cells, salt stress causes oxidative damage that destroys cellular macromolecules and compromises the integrity of cell membranes [14]. Oxidative stress biomarkers such as MDA, H_2_O_2_, and O_2_^−^ increase rapidly under high salinity conditions [15], and plants induce their own up-regulated enzymes (SOD, POD, CAT, and APX) and non-enzymatic antioxidants (AsA and GSH) to scavenge ROS and maintain ROS homeostasis [16].

Plants activate their self-defense mechanisms in response to unfavorable environmental circumstances, and one of the most important defense mechanisms is osmoregulation. Salt stress disrupts the ionic balance in plants and hinders normal physiological and biochemical processes [17,18]. Under salt stress, there was a gradual increase in sodium ion content and a decrease in potassium ion content in plants [16,19]. In addition, Na^+^ accumulation also causes osmotic stress, which is caused by salt stress and prevents plants from absorbing and utilizing water [20]. Salt stress reduces starch content and inhibits amylase activity, which in turn reduces the content of soluble sugars and the accumulation of osmotic substances [21,22].

Phytohormones are crucial regulators in plants for enhancing salt stress tolerance through various mechanisms such as reshaping ionic homeostasis, regulating the synthesis of osmoregulatory substances, and activating antioxidant enzymes [23]. Endogenous hormones play a vital role under salt stress as plants activate specific protective mechanisms in response to adversity, leading to changes in hormone levels that ultimately boost plant resistance. For instance, ABA is a key phytohormone involved in plant stress signaling and plays a critical role in controlling the expression of osmotic stress-responsive genes, thereby enhancing plant stress tolerance [24,25]. Research has demonstrated that the application of exogenous hormones can significantly enhance plant tolerance to stress. For example, treating plants with exogenous ABA improves stress tolerance [26,27], while IAA treatment reduces the negative effects of salt stress. In addition, stimulation of seeds with CTK improved salt tolerance in plants, increased BR signaling activity improved salt tolerance in plants [28], and exogenous GA reduced the effects of salt stress on plants, thereby improving plant resistance [29,30].

The application of exogenous regulators to promote plant salt tolerance is considered to be an effective way to improve crop resistance to salt stress [14,31]. Numerous studies have been conducted to explore the impact of exogenous substances on salt stress, focusing on their role in regulating cell membrane ionic homeostasis, inducing osmotic substance synthesis, enhancing antioxidant enzyme activity, promoting photosynthesis, regulating hormone signaling, and influencing gene expression [14]. Eugenol (C_10_H_12_O_2_, 4-allyl-2-methoxyphenol) is the primary component of clove essential oil [32], known for its clinical significance due to its anti-inflammatory and antibacterial properties [33]. As a natural bioactive compound, eugenol is utilized as a food preservative for food preservation purposes [34]. Eugenol inhibits the growth of several agricultural pathogens [35,36], which implies that eugenol has a great potential for use in agriculture. Studies have shown that eugenol increases plant tolerance to abiotic stress. Eugenol modulates antioxidant defence through hydrogen sulphide and abscisic acid-mediated signaling (e.g., heavy metal, drought, and cold), respectively, which confers drought tolerance to the tea tree and heavy metal tolerance to *Brassica juncea* [37,38]. Therefore, we speculate about the potential of eugenol to modulate the physiological functions of salt tolerance in plants. Investigating the ability and mechanisms by which eugenol confers salt tolerance in plants will help to expand the biological functions of eugenol in agriculture.

With the development of high-throughput sequencing technology and comparative genomics, transcriptome sequencing (RNA-seq) has become a popular method to study plant adversity mechanisms [39], and the screening of salt-tolerant genes lays the foundation for mining plant salt-tolerance mechanisms for further research. With the assembly and annotation of the tobacco genome, more and more RNA-seq has been used to investigate the plant adversity mechanism, for example, using it to explore the regulation of exogenous K^+^ on Ca^2+^ signaling and antioxidant activity as well as hormone transduction in tobacco under salt stress [40] and the regulation of exogenous melatonin on the root structure, antioxidant activity, and hormone levels in cotton under salt stress [41]. Salt stress seriously affects crop yield quality, threatening food security. How to alleviate salt stress has attracted the attention of a wide range of researchers. In this study, we exogenously applied eugenol under salt stress conditions and analyzed the effects of eugenol on the growth and development, oxidative damage, osmotic regulation, and hormone levels of tobacco seedlings under salt stress through physiology and transcriptomics to investigate the regulatory mechanism of eugenol on the salt tolerance of tobacco seedlings. As a result, we proposed the following hypothesis: (1) Eugenol alleviates the growth inhibition of tobacco seedlings by salt stress. (2) Eugenol mitigates the adverse effects of salt stress by increasing carbon and nitrogen metabolism, activating the antioxidant system, promoting osmoregulation, and coordinating hormone levels. (3) Eugenol induces up-regulation/down-regulation of genes under salt stress, thereby increasing salt tolerance in tobacco seedlings.

## 2. Results

### 2.1. Effect of Eugenol on the Growth and Development of Tobacco Seedlings under Salt Stress

Salt stress significantly reduced the biomass of tobacco seedlings, and exogenous eugenol mitigated this adverse effect. There was a concentration effect, with the best relief at 60 μM (Figure 1E,F). Compared with NaCl alone, 60 μM eugenol increased the fresh and dry weights of shoot tobacco seedlings under salt stress by 86.00% and 106.67%, respectively (Figure 1A,B), and the fresh and dry weights of the root increased by 100.00% and 100.00%, respectively (Figure 1C,D).

Salt stress significantly affected the root development of tobacco seedlings. The length, surface area, volume, average diameter, and number of root tips of seedlings were significantly reduced under salt stress, and the growth of the root system was inhibited. The root growth was restored after the application of exogenous eugenol, with 60 μM being the best restoration. Eugenol at 60 μM increased the length, surface area, volumetric, average diameter, and root tip of tobacco seedling roots under salt stress by 59.74%, 58.12%, 64.13%, 35.9%, and 48.47%, respectively, compared with NaCl treatment (Table 1).

### 2.2. Effect of Eugenol on Carbon and Nitrogen Metabolism of Tobacco Seedlings under Salt Stress

Salinity induces changes in the activities of enzymes involved in carbon and nitrogen metabolism. Compared with CK, NaCl treatment significantly reduced the activities of NR, SS, and GS in tobacco seedlings by 40.97%, 10.00%, and 5.80%, respectively. The activities of NR, SS, and GS in tobacco seedlings were significantly increased by NaCl with the addition of eugenol (Figure 2), by 31.03%, 5.80%, and 51.06%, respectively, but could not reach the level of CK.

### 2.3. Effect of Eugenol on the Antioxidant System of Tobacco Seedlings under Salt Stress

Salinity always induces the accumulation of ROS in plants. We measured the MDA content, which is a typical indicator of ROS-induced oxidative damage. The results showed that eugenol inhibited the NaCl-induced increase in the MDA content of seedlings, reducing it by 33.86% (Figure 3E). Compared with CK, the content of O_2_^−^ in tobacco seedlings was significantly increased after exposure to NaCl, increased by 94.1%, and the content of O_2_^−^ in NaCl-treated seedlings was significantly reduced by the addition of eugenol, reduced by 29.83% (Figure 3F).

Four typical antioxidant enzymes (SOD, POD, CAT, and APX) were measured in tobacco seedlings. The activities of all these enzymes were significantly enhanced by NaCl treatment compared to CK, increasing by 27.20, 202.56, 26.47, and 457.58%, respectively. The addition of eugenol further significantly increased the activities of these four enzymes compared to NaCl treatment alone, increasing by 24.38, 18.22, 21.60, and 28.8%, respectively (Figure 3A–D).

GSH is a strong, non-enzymatic antioxidant found in plants. The addition of eugenol significantly increased the content of GSH in tobacco seedlings compared with NaCl treatment alone, increasing by 29.49% (Figure 3G). GSSG is the oxidized form of GSH. The addition of eugenol significantly reduced the content of GSSG in tobacco seedlings compared with NaCl treatment alone, reduced by 17.89% (Figure 3H). In addition, salt stress significantly reduced the ratios of GSH/GSSG in NaCl-treated tobacco seedlings (Figure 3I), whereas the addition of eugenol significantly increased their ratios.

### 2.4. Effect of Eugenol on Osmoregulation in Tobacco Seedlings under Salt Stress

Inorganic ions are important components in and out of cells in plants, and ion homeostasis is important for plants to carry out normal physiological functions. When the influx of ions exceeds the efflux, salinity accumulates Na^+^ and Cl^−^ within the leaves. Treatment with NaCl alone significantly decreased the K^+^ content of tobacco seedlings by 35% and increased the Na^+^ content by 367.74% compared to CK. Exogenous eugenol increased the K^+^ content of tobacco seedlings under salt stress by 41.25% and decreased the Na^+^ content by 34.34% compared to NaCl treatment (Figure 4).

Soluble sugars and starch are important osmoprotectants for plants and play an important role in buffer regulation. Compared with CK, the soluble sugar and starch contents of tobacco seedlings treated with NaCl alone decreased by 17.03% and 23.09%, respectively, whereas exogenous eugenol increased the soluble sugar and starch contents of tobacco seedlings under salt stress by 7.72% and 25.42%, respectively (Figure 5).

### 2.5. Effect of Eugenol on Hormone Levels in Tobacco Seedlings under Salt Stress

The ABA content in tobacco seedlings under NaCl treatment was 76.89% higher than the control, while the ABA content in NaCl + eugenol treatment was 51.93% higher than NaCl (Figure 6A). In addition, the GA_3_ content in tobacco seedlings under NaCl treatment was 59.28% lower than the control, while the GA_3_ content in NaCl + eugenol treatment was 266.28% higher than that in tobacco seedlings under NaCl treatment (Figure 6B). There was no significant effect of ABA and GA_3_ content on tobacco seedlings treated with eugenol alone relative to the control.

### 2.6. Transcriptome Analysis

Three replicates of both NaCl and NaCl+Eu treatments were analyzed by RNA-seq technology and reference transcriptome sequencing. Over 35,691,234 reads were obtained from each sample by raw read collation and filtering. After filtering, more than 98.67% of the valid bases and more than 94.29% of the Q30-valued bases were obtained from each sample. The accuracy of the measured data was high, which helped in analyzing the data at a later stage.

Principal component analysis showed that the NaCl treatment was significantly different from the NaCl+Eu treatments, with similar within-group reproducibility for the three biological replicates; this includes 2.5% for PC1 and 97.5% for PC2. (Figure 7A). Differential gene expression analysis of all expressed genes identified a total of 2371 differentially expressed genes, of which, 1001 were up-regulated and 1370 were down-regulated (Figure 7B).

Gene ontology (GO) analyses of the differentially expressed genes showed significant differences between NaCl treatment and NaCl+Eu treatment in terms of biological processes, cellular composition, and molecular functions. Among them, in terms of molecular function, the top three were binding to tetrapyrrole, molecular function, and oxidoreductase activity; in terms of biological process, the top three were biological process, redox process, and photosynthesis; and in terms of cellular composition, the top three were extracellular region, cellular components, and photosystem I. The top three were the extracellular region, cellular components, and photosystem I (Figure 7C).

In order to deeply analyze the functions of the differentially expressed genes, we selected the most significant KEGG terms (*p* < 0.05), and the differentially expressed genes between treatments were subjected to KEGG enrichment analysis, which were mainly enriched in phenyopropanoid biosynthesis (Ko00940), MAPK signaling pathway-plant (Ko04016), phytohormone signaling (Ko04075), glutathione metabolism (Ko00480), and glyoxylate and dicarboxylic acid metabolism pathways (Ko00630) (Figure 7D).

#### 2.6.1. Antioxidant Enzyme System

Eugenol induced changes in antioxidant enzyme activities in tobacco seedlings under salt stress, and the transcriptional results showed that eugenol induced an increase in the expression of antioxidant enzyme-related genes, including a significant up-regulation of 4 SOD (LOC107819573-superoxide dismutase [Fe], LOC107820063-superoxide dismutase [Fe]3, LOC107767528-superoxide dismutase [Cu-Zn], LOC107790449-superoxide dismutase [Cu-Zn]), a significant up-regulation of 5 POD (LOC107759554-peroxidase 51-like, LOC107765883-peroxidase 63-like, LOC107770624-peroxidase 12-like, LOC107773042-peroxidase N1, LOC107780248-peroxidase 21-like) were significantly up-regulated, and 7 CAT (LOC107826360-catalase isozyme 1-like, LOC107759410-catalase isozyme 3-like, LOC107767654-catalase isozyme 3-like, LOC107771182-catalase isozyme 3, LOC107781013-catalase isozyme 3-like, LOC107786140-catalase isozyme 1) were significantly up-regulated, 6 APX (LOC107759703-L-ascorbate peroxidase 2, LOC107765289-L-ascorbate peroxidase 2, LOC107776943-L-ascorbate peroxidase 1, LOC107782355-L-ascorbate peroxidase 3, LOC107783197-L-ascorbate peroxidase 3) were significantly up-regulated, 1 CAT (LOC107828252-catalase isozyme 1-like) and 1 APX (LOC107775122-L-ascorbate peroxidase 3) were downregulated (Figure 8).

#### 2.6.2. Phytohormone Signaling

Eugenol altered hormone levels and induced phytohormone signaling in tobacco seedlings under salt stress (Ko04075). Among them, in the ABA synthesis pathway, eugenol induced a significant down-regulation of five PP2C (LOC107824451- serine/threonine-protein kinase SRK2E-like, LOC107785034, LOC107772435- protein kinase and PP2C-like domain-containing protein, LOC107765384-protein kinase and PP2C-like domain-containing protein, LOC107825832-protein kinase and PP2C-like domain-containing protein) and four SnRK2 (LOC107817827-serine/threonine-protein kinase SRK2A-like, LOC107792070-serine/threonine-protein kinase SRK2I-like, LOC107791496-serine/threonine-protein kinase SRK2A, LOC107766520-serine/threonine-protein kinase SRK2A-like) were significantly up-regulated (Figure 9A).

In the GA synthesis pathway, eugenol induced up-regulation of four GID1 (LOC107813435-gibberellin receptor GID1C-like, LOC107799051-S-type anion channel SLAH4-like, LOC107776835-gibberellin receptor GID1C-like, and LOC107831490-gibberellin receptor GID1B-like) under salt stress compared with NaCl treatment (Figure 9B).

In IAA metabolism, eugenol induced the up-regulation of 4 TIR1 (LOC107818988-protein TRANSPORT INHIBITOR RESPONSE 1-like, LOC107821218-protein TRANSPORT INHIBITOR RESPONSE 1-like, LOC107780463-protein TRANSPORT INHIBITOR RESPONSE 1-like, LOC107791833-protein TRANSPORT INHIBITOR RESPONSE 1-like), 23 AUX/IAA (LOC107820037-auxin-responsive protein IAA27-like, LOC107822266-auxin-responsive protein IAA4-like, LOC107823026-auxin-responsive protein IAA17-like, LOC107825977-auxin-responsive protein IAA8-like, LOC107830067-auxin-responsive protein IAA4-like, LOC107830653-auxin-responsive protein IAA14-like, LOC107762242-auxin-responsive protein IAA28-like, LOC107763299-auxin-responsive protein IAA8-like, LOC107767650-auxin-responsive protein IAA9-like, LOC107772558-auxin-responsive protein IAA13-like, LOC107774041-auxin-responsive protein IAA4-like, LOC107779361-auxin-responsive protein IAA20-like, LOC107783207-auxin-responsive protein IAA27-like, LOC107786124-auxin-responsive protein IAA13-like, LOC107786810-auxin-responsive protein IAA28-like, LOC107788572-auxin-responsive protein IAA8-like, LOC107789100-auxin-responsive protein IAA4-like, LOC107789101-auxin-responsive protein IAA27-like, LOC107794264-auxin-responsive protein IAA16-like, LOC107797525-auxin-responsive protein IAA26-like, LOC107813390-auxin-responsive protein IAA29-like, LOC107814232-auxin-responsive protein IAA27-like, LOC107815530-auxin-responsive protein IAA14-like) up, 5 GH3 (LOC107821957-putative indole-3-acetic acid-amido synthetase GH3.9, LOC107770881-indole-3-acetic acid-amido synthetase GH3.6-like, LOC107770902-indole-3-acetic acid-amido synthetase GH3.6-like, LOC107789037-probable indole-3-acetic acid-amido synthetase GH3.1, LOC107794279-probable indole-3-acetic acid-amido synthetase GH3.1) up, 1 AUX/IAA (LOC107773398-auxin-responsive protein IAA1-like) down, 1 GH3 (LOC107796246-indole-3-acetic acid-amido synthetase GH3.6-like) downward (Figure 9C).

## 3. Discussion

Increasing crop salt tolerance through the use of exogenous regulating chemicals is a straightforward and practical approach with a wide range of potential applications. Soil salinity has emerged as one of the most detrimental issues facing agricultural land. Eugenol is a natural compound extracted from plants that has a wide range of antioxidant, antibacterial, and anti-inflammatory properties [42], but its use in improving plant salt tolerance has rarely been reported. In the study, eugenol was found to increase antioxidant activity and content and harmonize ionic balance and hormone levels, thereby mitigating the adverse effects of salt stress on tobacco seedlings. In order to explore the adaptation response to salt stress mediated by eugenol, we performed comparative transcriptome analysis based on the RNA sequences of salt treatment and salt treatment + eugenol and analyzed the two control transcriptional differences between the treatments.

Carbon and nitrogen metabolisms are essential processes in plant life [9]. Nitrogen is one of the important mineral nutrients required by plants [43], and GS and NR play a key role in nitrogen uptake and utilization, particularly in relation to ammonium [44]. Current studies have found that salt stress affects plant uptake and assimilation of nitrogen by affecting nitrogen metabolism enzyme activities [12,13]. Carbon is also an important component of plant biomass [45], and SS mainly regulates starch synthesis as well as sucrose degradation processes, affecting the carbohydrate level of plants [46]. In this study, we found that the activities of NR, GS, and SS were significantly reduced in tobacco seedlings under salt stress, which also indicated that salt stress inhibited the activities of enzymes related to carbon and nitrogen processes, and reduced nitrogen uptake and utilization as well as carbohydrate synthesis in tobacco seedlings, which is in agreement with the results of previous studies [13,47]. In contrast, exogenous eugenol increased the activities of these enzymes under salt stress and promoted carbon and nitrogen metabolism. It was shown that eugenol could increase the activities of enzymes related to carbon and nitrogen metabolism and ameliorate the inhibitory effect of salt stress on the processes of carbon and nitrogen metabolism in plants.

For plants to grow optimally, ROS production is essential. When exposed to abiotic stress, plants experience a rapid increase in ROS levels within their cells, leading to detrimental effects such as protein denaturation, DNA mutation, and membrane lipid peroxidation [48]. ROS are highly reactive molecules that are rapidly scavenged or degraded and are highly sensitive to any environmental changes, making them very unstable and difficult to detect directly [49]. O_2_^−^ is often considered the first ROS generated in plant cells upon stress conditions, and with a short half-life (a few milliseconds at neutral pH), it is a single-electron reduction product of oxygen molecules [50]. Due to its low short-term survival and high activity, it is usually detected by indirect methods, such as the hydroxylamine oxidation reaction [51]. Salt stress often leads to the accumulation of ROS, resulting in oxidative damage in plants, a phenomenon that was also noted in our current research.

SOD is the first line of defense for ROS scavenging and can catalyze O_2_^−^ to H_2_O_2_ and O_2_, while POD, CAT, APX, and GSH can further catalyze H_2_O_2_ to H_2_O [52,53]. MDA indicates the level of cytoplasmic peroxidation and the strength of the response to adverse conditions. Salt stress increased the elevated MDA content in the cells, which indicates oxidative damage to the cells. Tobacco seedlings maintain normal cellular physiological and metabolic processes by initiating antioxidant systems, including enzymatic and non-enzymatic defenses, to reduce the toxic effects of excess ROS. This study found that salt stress resulted in a significant increase in the activity of these enzymes. However, the increase in these enzymes may not be enough to scavenge salt induced large amounts of ROS. The addition of eugenol further increased the activities of these enzymes, which further significantly reduced the ROS levels in tobacco seedlings under salt stress and maintained the redox balance of plant cells, which may be related to the up-regulation of antioxidant enzyme-related genes in the transcriptional results of NaCl + eugenol treatment.

In this situation, the plant defense system responds rapidly to counteract the damage of salt stress through a range of salt tolerance mechanisms [54]. Plants utilize enzymatic or non-enzymatic antioxidant induction as a key strategy to combat ROS stress caused by salt [55], as it is thought to be an efficient way to delay the damaging effects of adversity on cells and tissues [55,56]. GSH is a ubiquitous tripeptide involved in the transduction of environmental signals and plays a key role in plant responses to abiotic stresses. GSH is a strong non-enzymatic antioxidant found in plants that can interact with growth factors to modulate plant tolerance under stress conditions [56]. GSH can undergo reversible oxidation to form GSSG. GSH/GSSG is one of the most important redox pairs in cells. Therefore, the determination of intracellular GSH and GSSG content and the GSH/GSSG ratio can well reflect the redox state of the cell. The results of the present study showed that eugenol increased the GSH content of tobacco seedlings under salt treatment and contributed to the scavenging of reactive oxygen species. In addition, eugenol significantly increased GSH/GSSG in salt-treated tobacco seedlings, suggesting that eugenol has the ability to maintain the cellular reductive state against ROS-induced oxidative stress.

When plants are subjected to salt stress, a complex series of osmotic stresses and osmotic responses occur [57], Salt stress can result in the entry of large amounts of Na^+^ into plant cells, and the excessive accumulation of Na^+^ is toxic to cytoplasmic enzymes, leading to ionic stress. Plants regulate the internal concentration of Na^+^ by chelating it into vesicles or expelling it from cytoplasmic solutes. The process of osmoregulation, which controls osmotic substances and inorganic ions to maintain cellular water potential, is necessary for the plant to continue performing its life functions. Thymol inhibits Na^+^ inward flow and reduces Na^+^ in tobacco seedlings through the activation of *NtSOS1* and *NtNHX1* [58]. Our study revealed an increase in Na^+^ accumulation and a decrease in K^+^ content in the leaves of salt-treated tobacco seedlings. This could be attributed to Na^+^ accumulation hindering K^+^ uptake by the tobacco seedlings’ root system. Exogenous eugenol was effective in boosting K^+^ content and curbing excessive Na^+^ accumulation. This also suggests that eugenol can reduce the excessive accumulation of Na^+^ in plant phytoplankton, maintain the cellular water potential, and maintain the integrity of the cell membrane, but the mechanism of its regulation of ionic signaling needs to be further investigated.

The buildup of sugars has osmotic effects, protects against osmotic pressure, regulates osmoregulation, scavenges free radicals, and provides membrane protection against salt stress. Its content also indicates how resistant the plant is to abiotic stressors [9,12,13,59]. In this study, both starch and soluble sugars decreased significantly in tobacco seedlings under salt stress, which is consistent with previous studies [60]. This reduction is likely due to salt stress causing cell membrane damage, leading to the permeation of soluble substances. In contrast, exogenous eugenol effectively increased the accumulation of starch and soluble sugars, which may be related to the up-regulation of hashish kinase (HK), which mediates sugar sensing and hashish phosphorylation, and its accumulation protects the cell membrane, reduces the leakage of osmotic substances, and increases the osmoregulatory capacity as well as the scavenging of free radicals in tobacco seedlings. These results suggest that eugenol has the potential to enhance osmotic substance accumulation in response to salt stress injury.

Phytohormones mediate plant responses to abiotic stresses. Previous studies have shown that ABA and GA are classical phytohormones that regulate plant growth and abiotic stress and that the salt tolerance of plants can be improved by regulating the levels of ABA and GA [20,23]. In tomatoes, ABA content increased significantly and GA level decreased under salt stress [61]. Exogenous ABA induced the levels of ABA and indoleacetic acid in Acacia bicolor seedlings under salt stress, and transcriptional analyses showed that the differentially expressed genes were involved in metabolic pathways such as phytohormone signaling and proline metabolism [62].In this study, we found that exogenous eugenol significantly increased endogenous ABA and GA_3_ levels in tobacco seedlings under salt stress, and further RNA-seq analyses showed that DEGs were significantly enriched in phytohormone signaling, with changes in the expression of genes related to ABA and GA signaling pathways. In ABA regulation, eugenol induced the expression of PTR/PYL under salt stress, further repressed the expression of PP2C, and up-regulated the expression of SnRK2, thereby increasing ABA levels in response to salt stress. In GA regulation, exogenous eugenol upregulated GID1 (a GA receptor) in the signaling pathway, thereby increasing GA_3_ levels. These results also suggest that eugenol can mediate phytohormonal responses to abiotic stress by regulating the expression of genes involved in the ABA and GA signaling pathways, thereby increasing ABA and GA_3_ levels in plants.

## 4. Materials and Methods

### 4.1. Plant Culture and Treatment

The seeds of tobacco (K326) were provided by the Germplasm Resources Laboratory of Shandong Agricultural University. Seedlings were nursed in the greenhouse of Shandong Agricultural University by a floating nursery, and 45 days after seeds were sown, uniformly growing and healthy seedings, were selected and used for hydroponics. After two days of adaptation, seedlings were transferred into the Hoagland nutrient solution, different eugenol concentrations (30, 60, 90 μM) were set, from which the optimum concentration was screened for subsequent experiments, and the seedlings were treated according to CK (Hoagland’s), NaCl (Hoagland’s + 150 mM NaCl), NaCl+Eu (Hoagland’s + 150 mM NaCl + 60 μM Eu), and Eu (Hoagland’s + 60 μM Eu) for seedlings, with the nutrient solution changed once in 2–3 days, pH 6.5–7.0, and continuous aeration. Seedlings were grown under the following conditions: incubated in a plant growth cabinet with a light intensity of 5000 lx (air relative humidity of 50%, photoperiod of 12 h, and temperature of 26 °C), and relevant physicochemical properties and transcriptome assays were performed after 15 days.

### 4.2. Measurement of Biomass

The above-ground and below-ground parts of tobacco were weighed on an electronic balance to obtain the fresh weight of tobacco seedlings. Afterwards, the seedling parts were put into a blast drying oven, killed at 105 °C for half an hour, dried at 80 °C and then the dry weight of the seedlings was determined.

### 4.3. Root System Parameters

Total root length, total surface area, total volume, average diameter, and number of root tips of tobacco seedlings were determined using a root scanner (ScanMaker i800plus, Hangzhou Wanshen Inspection Technology Co., Hangzhou, China).

### 4.4. Carbon and Nitrogen Metabolism

About 0.1 g of fresh seedlings were taken, and after grinding, nitrite reductase (NR) activity was determined based on NR-catalyzed reduction of nitrate to nitrite with a characteristic absorption peak of NADH at 340 nm (BC0080, Solarbio, Beijing, China).

About 0.1 g of fresh seedlings were taken, and after grinding, the sucrose synthase (SS) activity was determined on the basis of SS- catalyzed reaction of free fructose with glucose donor UDPG to form sucrose, which reacts with resorcinol to show a color change with a characteristic absorption peak at 480 nm (BC0580, Solarbio, Beijing, China).

About 0.1 g of fresh seedlings were taken, and after grinding, glutamine sythetase (GS) activity was determined based on the synthesis of glutamine from ammonium ions and glutamic acid catalyzed by GS in the presence of ATP and Mg^2+^; glutamine was further converted to γ-glutamyl isohydroxamic acid, which formed a reddish complex with iron in the presence of acid; the complex had a maximum absorption peak at 540 nm (BC0910, Solarbio, Beijing, China).

### 4.5. ROS Measurement

About 0.1 g of fresh seedlings were taken, and after grinding, MDA condensed with thiobarbituric acid (TBA) under acidic and high temperature conditions to form the brownish-red color of trimethoprim (3,5,5-trimethyloxazole-2,4-dione), which has a maximum absorption wavelength of 532 nm. Colorimetric analysis can be used to estimate the amount of MDA in a sample (BC0025, Solarbio, Beijing, China).

About 0.1 g of fresh seedlings were taken, and after grinding, superoxide anion reacts with hydroxylamine hydrochloride to form NO_2_^−^, NO_2_^−^ in the presence of paminobenzenesulfonamide and naphthalene ethylenediamine hydrochloride to form a purplish-red azo compound with a characteristic absorption peak at 530 nm, and the O_2_^−^ content in the samples can be calculated from the A530 value (BC1290, Solarbio, Beijing, China).

### 4.6. Assay of the Activity of Antioxidative Enzymes

#### 4.6.1. Antioxidant Enzymes

About 0.1 g of fresh seedling samples were homogenized with 2 mL of cold phosphate buffer (50 mM, pH 7.0). Then the mixture was centrifuged at 12,000× *g* for 15 min (4 °C). The supernatant was collected for the determination of enzymatic activity. The SOD activity was determined spectrophotometrically (OD560 nm) using a commercial kit (BC0170, Solarbio, Beijng, China) based on the quantification of the inhibition of the photochemical reaction of NBT (nitro-blue tetrazolium) in a reaction system with methionine and riboflavin. The CAT activity was determined spectrophotometrically (OD240 nm) using a commercial kit (BC0205, Solarbio, Beijng, China) based on the decomposition of H_2_O_2_. The POD activity was determined spectrophotometrically (OD470 nm) using a commercial kit (BC0090, Solarbio, Beijing, China) based on the oxidation rate of guaiacol in the presence of H_2_O_2_. The APX activity was determined spectrophotometrically (OD290 nm) using a commercial kit (BC0220, Solarbio, Beijing, China) based on the oxidation rate of AsA in the presence of H_2_O_2_.

#### 4.6.2. Antioxidant Content

GSH content was determined using a commercial kit (BC1170, Solarbio, Beijing, China) based on the spectrophotometric determination of the reaction product (OD265 nm) of glutathione with DTNB (5,5′-dithiobis-2-nitroenoic acid). Using a commercial kit (BC1180, Solarbio, Beijing, China), the content of GSSG was determined based on the rate of GSSG production from GSH catalyzed by GR (GSH reductase) (measuring OD265 nm for the reaction of GSH with DTNB).

### 4.7. Measurement of Hormone Content

The contents of ABA and GA_3_ were determined by Shandong Guocangjian Biology Co. (Taian, China).

### 4.8. Measurement of the Osmoregulator

Dried seedling samples were digested with H_2_SO_4_ (containing 10% H_2_O_2_) using a microwave digestion system (MARS6, CEM Corporation, Matthews, NC, USA). Na^+^ and K^+^ were then determined by atomic absorption spectrometry (PinAAcle 900T, PerkinElmer, Hopkinton, MA, USA).

Application of the Solarbio Commercial Kit BC0035 (Beijing, China) is based on the determination of plant soluble sugar content by anthrone colorimetry (OD620nm). Application of the Solebo commercial kit BC0700 is based on the separation of starch with 80% ethanol and acid hydrolysis to glucose, followed by the determination of starch content by anthrone colorimetry (OD620nm).

### 4.9. Transcriptome Analysis

Total RNA was extracted, concentration and purity were detected by nanodrops, RNA integrity and DNA contamination were assessed by agarose gel electrophoresis, and RNA integrity was accurately detected by using the Agilent 2100 Bioanalyzer.RNA 6000 Nano kit 5067-151 (Agilent, Santa Clara, CA, USA). The Agilent High Sensitivity DNA Kit (Agilent Technologies Inc., Santa Clara, CA, USA, 5067-4626) for library quality testing. The mixed libraries were diluted, quantified stepwise, and sequenced in PE150 mode on an Illumina sequencer.

HTSeq (v0.9.1) was used to statistically compare the read count value of each gene, which was used as the raw expression level of the gene. In order to make the gene expression levels of different genes and samples comparable, FPKM was used to normalize the expression (normalization). *p*-value (Padj) < 0.05 and |log2(-FoldChange)| ≥ 1 were regarded as differentially expressed genes. RNA differential expression analysis between the two groups was performed using DESeq2 software(v.1.38.3).

GO enrichment analysis was performed using topGO (v2.50.0) to identify GO terms that were significantly enriched for differential genes (all/up/down) by calculating the *p*-value through a hypergeometric distribution method (*p*-value < 0.05 was used as the criterion for significant enrichment), thus identifying the main biological functions exercised by the differential genes. KEGG pathway enrichment analysis was performed using clusterProfiler (v4.6.0) software, focusing on significantly enriched pathways with a *p*-value < 0.05. Gene set enrichment analysis was performed using GSEA (v4.1.0) software. GSEA does not require the specification of an explicit differential gene threshold, ranks all genes according to the degree of differential expression in the two sets of samples, and then employs a statistical method to test whether a predefined set of genes is enriched at the top or the lower part of the ranked list.

### 4.10. Data Analysis

Raw data were subjected to an ANOVA (one-way analysis of variance) test followed by an LSD (Least Significant Difference) test in order to analyze significant differences of *p* < 0.05 between treatments (SPSS 25.0). Graphing was performed using Origin2024.

## 5. Conclusions

This experiment explored the ways in which eugenol improves salt tolerance in tobacco (Figure 10). Firstly, exogenous eugenol alleviated the growth inhibition of tobacco seedlings by salt stress. Secondly, exogenous eugenol promoted carbon and nitrogen metabolism, activated the antioxidant system to reduce the accumulation of reactive oxygen species, facilitated osmoregulation, and coordinated hormone levels. Thirdly, eugenol alleviated salt stress by up-/down-regulating gene expression. These results would help develop new biostimulants for crop production in saline areas, which is important for both applied and fundamental research.

## Figures and Tables

**Figure 1 ijms-25-06771-f001:**
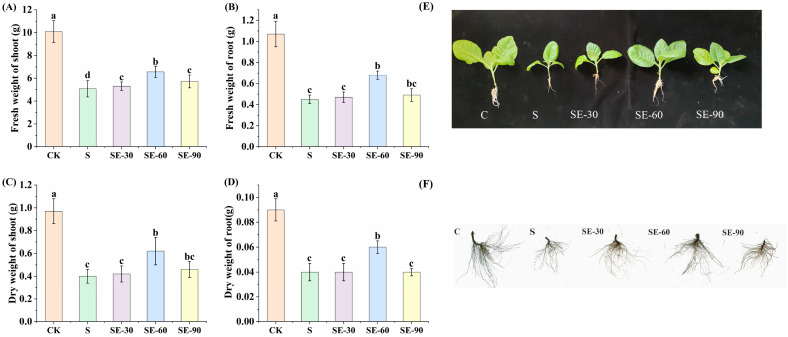
Effect of eugenol on the growth of tobacco seedlings under salt stress. (**A**) Fresh weight of the shoot. (**B**) Fresh weight of the shoot. (**C**) Dry weight of the root. (**D**) Dry weight of the root. (**E**) Phenotypic effects of different concentrations of eugenol on the growth of tobacco seedlings. (**F**) Effects of different concentrations of eugenol on the root system of tobacco seedlings. Different lowercase letters indicate significant differences between treatments (*p* < 0.05, LSD).

**Figure 2 ijms-25-06771-f002:**
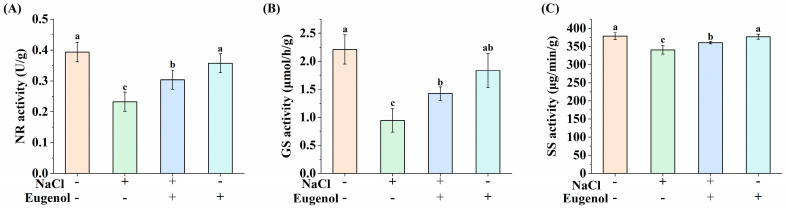
Effect of eugenol on carbon and nitrogen metabolism in tobacco seedlings under salt stress. (**A**) NR activity. (**B**) GS activity. (**C**) SS activity. Different lowercase letters indicate significant differences between treatments (*p* < 0.05, LSD).

**Figure 3 ijms-25-06771-f003:**
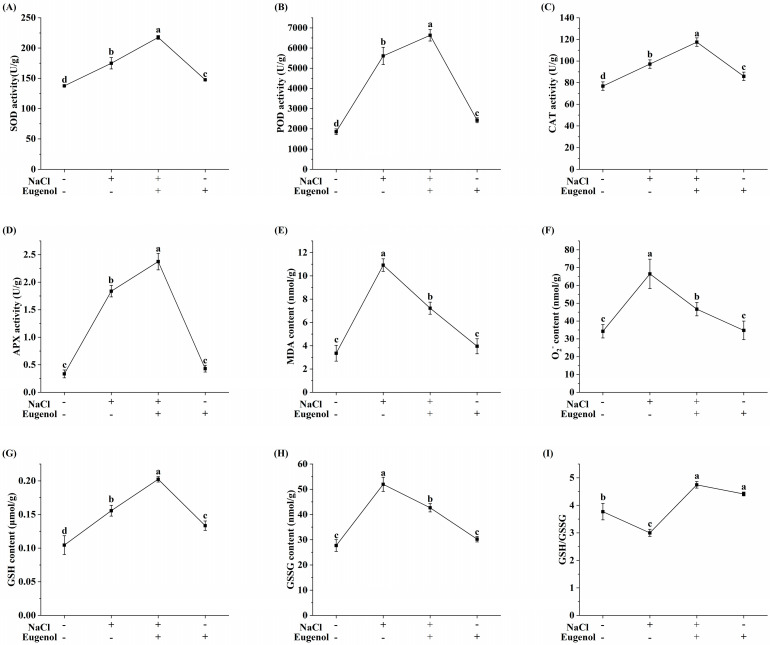
Effect of eugenol on the antioxidant system of tobacco seedlings under salt stress. (**A**) SOD activity. (**B**) POD activity. (**C**) CAT activity. (**D**) APX activity. (**E**) MDA content. (**F**) O_2_^−^ content. (**G**) GSH content. (**H**) GSSG content. (**I**) GSH to GSSG ratio. Different lowercase letters indicate significant differences between treatments (*p* < 0.05, LSD).

**Figure 4 ijms-25-06771-f004:**
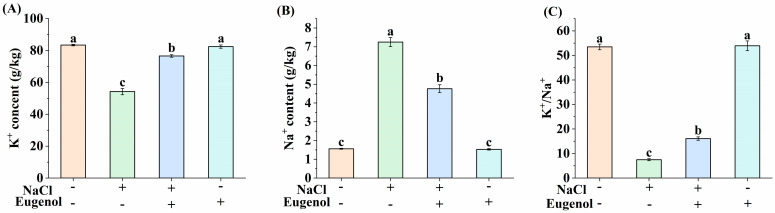
Effect of eugenol on the sodium and potassium content of tobacco seedlings under salt stress. (**A**) K^+^ content. (**B**) Na^+^ content. (**C**) K^+^/Na^+^. Different lowercase letters indicate significant differences between treatments (*p* < 0.05, LSD).

**Figure 5 ijms-25-06771-f005:**
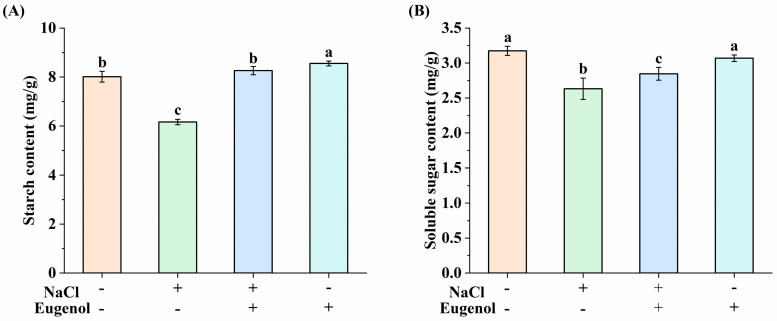
Effect of eugenol on osmotic substances in tobacco seedlings under salt stress. (**A**) Starch content. (**B**) Soluble sugar content. Different lowercase letters indicate significant differences between treatments (*p* < 0.05, LSD).

**Figure 6 ijms-25-06771-f006:**
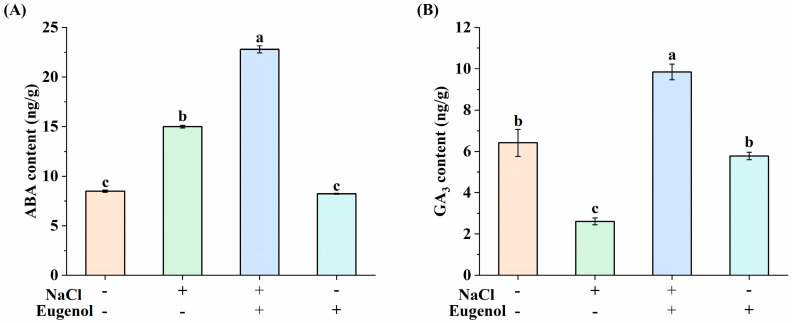
Effect of eugenol on the hormone contents of tobacco seedlings under salt stress. (**A**) ABA content. (**B**) GA3 content. Different lowercase letters indicate significant differences between treatments (*p* < 0.05, LSD).

**Figure 7 ijms-25-06771-f007:**
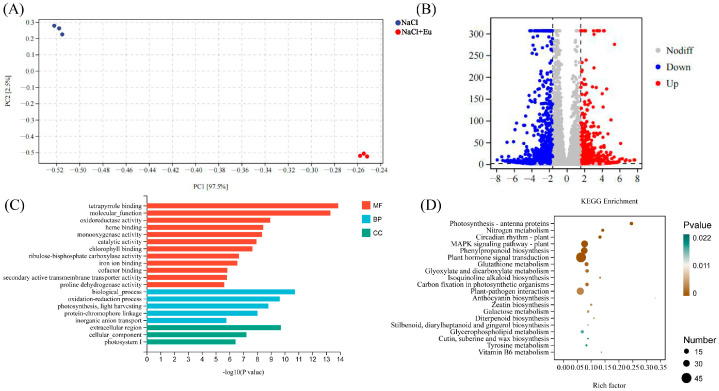
Analysis of differentially expressed genes (DEGs) between NaCl and NaCl+Eu. (**A**) Principal component analysis. (**B**) Volcano plots depicting up-regulated, down-regulated, and non-regulated genes between the two treatments. (**C**) GO enrichment analysis of DEGs. (**D**) KEGG enrichment analysis of DEGs.

**Figure 8 ijms-25-06771-f008:**
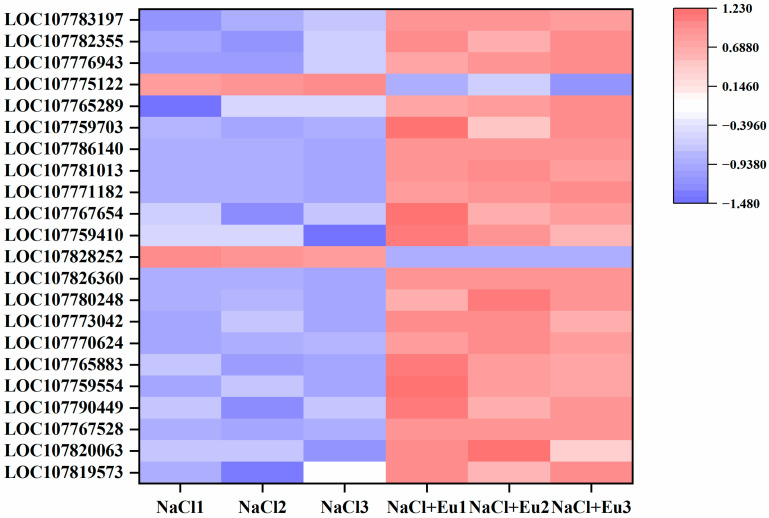
Effect of eugenol on the expression of antioxidant enzyme-related genes under salt stress.

**Figure 9 ijms-25-06771-f009:**
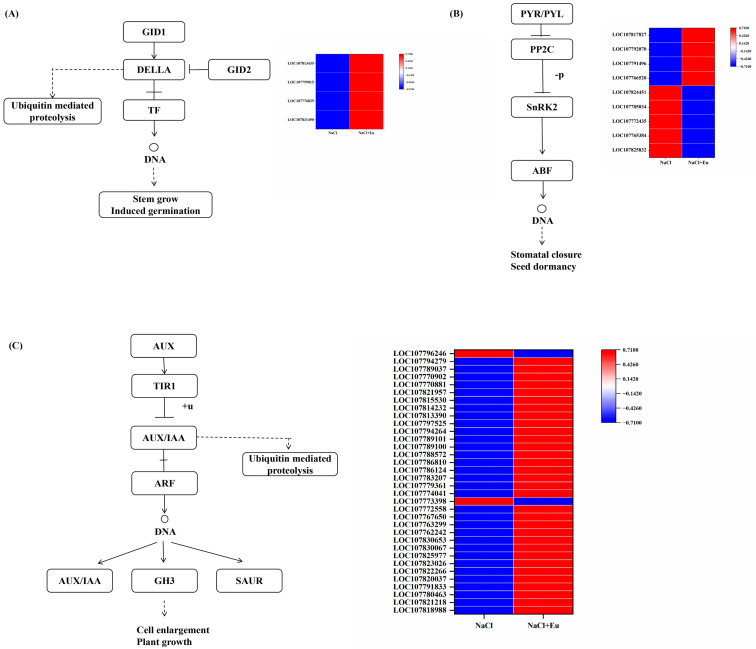
Effect of eugenol on the expression of genes involved in hormone metabolism in tobacco seedlings under salt stress. (**A**) Effect of eugenol on the expression of ABA-related genes under salt stress. (**B**) Effect of eugenol on the expression of GA-related genes under salt stress. (**C**) Effect of eugenol on the expression of IAA-related genes under salt stress.

**Figure 10 ijms-25-06771-f010:**
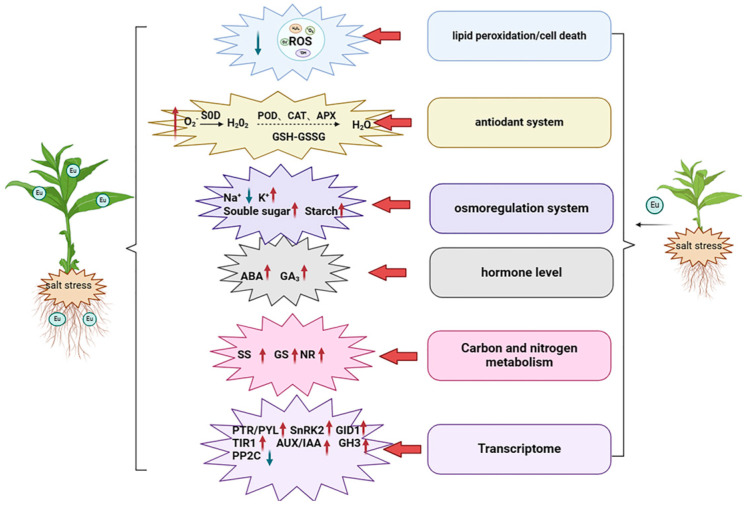
Schematic representation of the strategy of eugenol to alleviate salt stress in tobacco seedlings.

**Table 1 ijms-25-06771-t001:** Effect of exogenous eugenol on the root growth of tobacco seedlings. Different lowercase letters indicate significant differences between treatments (*p* < 0.05, LSD).

Treatment	Length (cm)	Surface Area (cm^2^)	Volumetric (cm^3^)	Average Diameter (mm)	Root Tip
CK	804.8 ± 65.87 a	169.22 ± 11.35 a	1.86 ± 0.29 a	0.67 ± 0.08 a	1499 ± 155 a
S	473.4 ± 29.14 c	103.11 ± 19.61 b	0.92 ± 0.11 c	0.39 ± 0.08 c	687 ± 60 c
SE-30	579.02 ± 32.86 b	121.97 ± 4.77 b	1.00 ± 0.14 c	0.51 ± 0.04 bc	910 ± 141 b
SE-60	656.21 ± 54.2 b	133.04 ± 13.12 b	1.31 ± 0.06 b	0.53 ± 0.07 b	1020 ± 100 b
SE-90	602.36 ± 53 b	118.29 ± 22.15 b	1.08 ± 0.11 c	0.50 ± 0.05 bc	955 ± 100 b

## Data Availability

Data is contained within the article.

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
