# Peer review of "Exogenous Eugenol Alleviates Salt Stress in Tobacco Seedlings by Regulating the Antioxidant System and Hormone Signaling"

_ijms, 2024, doi:10.3390/ijms25126771_

Round 1

Reviewer 1 Report

Comments and Suggestions for Authors

In the current paper authors combined treatemnt and exogenous eugenol with slat stress to evaluate it protective effect on tobacco seedlings.

Despite topic is interesting and actual, some conclusións required revisión and clarification.

One of the most interesting finding of the paper is extensión of superoxide life time. Authiors need to discuss this new funding in details.

Title is not correct. Physiology and transcriptomics itself can not reveal a mechanism.

Line 10-13: very clomplicated and long sentence, Please, Split and clarify.

Line 13: “small molecule compound” ¿  Compound is redundant.

Line 15_ seedling can not investigated by physiology and transcriptomis.

Line 19: “hormone level can not be improved.

While introduction provide comprehensive description of the topic, the organisation require significant improvement to avoid very common statemnet, more logic, removing repetition.

Line 56: plants do not fix nitrogen (except legumes). Here you shoukld write “metabolise”.

Line 62: Na itself can not cause damage. It cause vacuole alkalinization in some cell types.

Line 79 – 98: please, clarify exo and endogenous hormones, importance of local hormone distribution for stress responce, epigenetic etc.

Line 110: Abiotic stress can not have a physiological function.

Results: Line 134: fresh dry weight???

Lines 134- 139_ please, clarify numbers in the text and linked it with results.

Lines 166 – 167: interesting, that in our local conditions superxide half-life is so so liong.

 Lines 176 – 182: GSH has a low contribution to ROS scavenger.

 It has another mechansim of action. https://doi.org/10.3390/biom10111550

Figure 3D: which APX do you mean? cytAPX. Tsroma APX? Thylakoid APX? Peroxisomal APX? Each have a different meaning.

Figure 6ª: higher ABA should have an inhibitory effect, please, explain . Moreover, hormone act on organ/celular level and increasing/decreasing contents in different organs have a different meaning. Please, clarify organs/cell types.

Line 229: “from both NaCl+Eu treatment” ¿??

Line  : phenylpropanone biosynthesis may be phenyopropanoid?

Line 265: “Metabolismo hormonal” ¿?

Line 295: “ In this study, we identified the role of eugenol in mitigating salt stress, which enhances the antioxidant activity and content, and coordinates ion homeostasis and hormone levels to mitigate salt stress in tobacco seedlings under salt stress.” ¿???

Line 319: “Oxidative stress is essential.” ¿??? Not stress, but ROS production.

Figure 10: “Mechanism of exogenous eugenol to improve salt tolerance in tobacco seedlings”??? Do you mean eugenol applications? Which type of the cell do you nmean in this model? Each type of the cell has own mecahnism. There is common one!

Line 384: 45 days is not a seedlings, it should be plants.

Line 386: “retardation” = adaptation.

Line 394 “ relevant physicochemical properties” ¿??

Line 402: “total surface area, total volume,” can not be determined by scanner.

Line 406: grammar.

Lines 416 – 418: authors need to provode details how they able to extend superoxide life time from few milliseconds to hour. It is very inetersting form chemistry point of view.

Line 440: which part of the plants has been used for RNA ¿ Or you mix all rNA type in one preparation?

Line 446: words repetition.

Comments on the Quality of English Language

Many long sentences, some words repetitions, punctuations. Logic.

Reviewer 2 Report

Comments and Suggestions for Authors

The article presents the possibility of using eugenol to limit salt stress in tobacco cultivation. The authors used appropriate research methods. They presented the research results correctly in the form of charts and drawings. They based their discussion on the current literature on the problem in question. They interpreted the conclusions correctly.

Some shortcomings of the work include the lack of a research hypothesis, which should be added in the introduction.

After minor corrections, the work can be published in IJMS

Comments on the Quality of English Language

The English language is readable.

Reviewer 3 Report

Comments and Suggestions for Authors

The manuscript concerns the assessment of tobacco response to salt stress and the effect of eugenol in the mitigating the adverse effect of salt stress expressed in tobacco physiology and metabolism. The paper presents interesting data, but it has also some flaws that need to be corrected. The details are listed below:

Abstract:

L13: molecule or compound

L17-20: indicate some % changes of examined parameters between treatments

Introduction:

L31-32: briefly describe why the area of saline soil is growing

L68-98: shorten these paragraphs

L1-6-107: rephrase

L113: species name in italics

Results:

L140: check the letters in the Figure. Some values seem to be insignificant, while different letters are marked

L150: link the percentages to the exact examined parameter

L155-157: indicate values or % changes of examined parameters between treatments

L156: concentration of eugenol. Indicate that selected concentration of eugenol was chosen for further analysis

L164-182: indicate values or % changes of examined parameters between treatments

L247: what are A and B in Fig. 7A. Add correlations between parameters obtained in PCA

L253-283: gene names should be in italics

L263: replace the accession numbers by the names of genes encoding antioxidant enzymes

L265: rephrase

L266-283: add full names of genes

Discussion:

Discussion is mainly a summary of the results. Compare your own results with other studies with focus on the role of eugenol in mitigating salt stress or other abiotic stresses (if possible)

L319: ‘For plants to grow correctly, oxidative stress is essential’ - ??

L323-331: emphasize the ubiquitous role of antioxidant enzymes in the mitigating different abiotic stresses (salinity, pesticides, heavy metals, etc.). For this purpose refer to: https://doi.org/10.1016/j.chemosphere.2022.136284

Materials and Methods:

Briefly describe the analysis of antioxidant enzymes, antioxidant content, carbon and nitrogen metabolism, hormone content. Statement that they were analysed according to the manufacturer is insufficient.

L386: indicate the composition of the solution

L406: rephrase this statement in all paragraphs in Materials and Methods

L407-427: add full names for examined enzymes and non-enzymatic antioxidants

Comments on the Quality of English Language

Minor editing of English language required

Round 2

Reviewer 1 Report

Comments and Suggestions for Authors

Thank you very much for constructive corrections.

 Despite paper became better, some more corrections are required.

I mentioned some points, but authors need to carefully go through text again to find all potentail mistakes.

Title:

what is hormone level?

Phytohormones act locally and temporary, It is better to tell hormonal signaling or metabolism.

Line 13: properties – twice in one sentence.

Line 17: abbreviations is not acceptable in the abstracts without explanations.

Lines 16 – 26: a lot information without structure. Do you mean SOD activity and GSH contents, for example?

Line 86: osmotic pressure synthesis? Check other formulation as well.

Lines 95 – 98: require editing.

Line 103: regulating hormones?

Lines 110 – 114: split to two sentences.

Line 142: increased to what??

Lines 172 – 175: this is a very new funding and completely new definition as “superoxide level”. Authors need to discuss in details this funding about superoxide (authors seems to be extend life-time to few hours instead milliseconds). dx.doi.org/10.17504/protocols.io.bx49pqz6

Line 185: GSH is a regulator of auxin signaling, indeed (https://doi.org/10.3390/biom10111550).

Lines 333-334: here you described “molecular adaptation”. Response to stress generated after stress exposure in few minutes/hours. In your case you described adaptation mechanism and effect of eugenol in it.

Line 353: what is growth correctly?

Maybe optimal?

Lines 378 – 380: one of the mechanisms of Na effect is rapid uptake to vacuole and preventing to acidification ea. Preventing plant cell expansion. This can be mention in details as well.

Lines 467- 472: this is a most important funding of the paper, indeed! Authors, for the first time, demonstrated that superoxide is a very long living molecule. Authors need to discuss this funding in details and mentioned in the abstracts. Before authors protocol superoxide half-life has been detected as few milliseconds, but authors seem to be able to isolate and measure this molecule.

Line 491: “glutamine synthetase (GSSG) content based.” ???? Please, double check this part and make corrections.

Line 507: measurement = analysis

Comments on the Quality of English Language

some very long sentences, some problem with punctuations.

Reviewer 3 Report

Comments and Suggestions for Authors

The paper has been improved. I have one comment:

L16-24: divide this part on several sentences for better reading

Round 3

Reviewer 1 Report

Comments and Suggestions for Authors

Thank you! It seems all OK, except figure 3F. And main funding about superoxide half-life did not discussed. This need to be add before acceptance. Superoxide produced during impaired electron and degrate after few miiliseconds. The protocol authors used claimed that superoxide exist in extract after long long time, what mean automatically that superoxide still remian in it. It is a very new and must be discussed! dx.doi.org/10.17504/protocols.io.bx49pqz6

Comments on the Quality of English Language

minor polishing.

Author Response

Comments1: Thank you! It seems all OK, except figure 3F. And main funding about superoxide half-life did not discussed. This need to be add before acceptance. Superoxide produced during impaired electron and degrate after few miiliseconds. The protocol authors used claimed that superoxide exist in extract after long long time, what mean automatically that superoxide still remian in it. It is a very new and must be discussed! dx.doi.org/10.17504/protocols.io.bx49pqz6

Response1: We’re sorry for the misunderstanding for the measurement of superoxide radical. The half-life of superoxide radical is pretty short. It cannot exists for a long time enough to be extracted. We used hydroxylamine hydrochloride-based assay, a classical method, to quantify superoxide radical. The superoxide radical exists in plants under treatment. The fresh plant sample were homogenized with extraction buffer containing hydroxylamine hydrochloride. So as the plant cells broken, the superoxide radical reacted with hydroxylamine hydrochloride immediately to form NO2-. Then NO2- can be stable in the solution for the following reaction with paminobenzene-sulfonamide and naphthalene ethylenediamine hydrochloride to form the products that can be detected at OD 530 nm. . Thank you for pointing this out. We agree with this comment. Therefore, we have changed it on pages 11-12, lines 356-377 and page 18, lines 699-704.

The following references have been added:

  1. Pasternak, T.P.; Perez-Perez M. Optimization of ROS measurement and localization in plant tissues: challenges and solutions. protocols.io 2021, https://dx.doi.org/10.17504/protocols.io.bx49pqz6
  2. Zulfugarov, I.S.; Tovuu, A.; Kim, J.H.; Lee, C.H. Detection of Reactive Oxygen Species in Higher Plants. J. Plant Biol. 2011, 54, 351-357, doi:10.1007/s12374-011-9177-4.
  3. Elstner, F.; Heupel,A. Inhibition of nitrite formation from hydroxylammoniumchloride: A simple assay for superoxide dismutase. Analytical Biochemistry, 1976,70(2): 616-620. doi: 10.1016/0003-2697(76)90488-7.
